# New Coronavirus in Colombian Caribbean Bats: In Silico Analysis Reveals Possible Risk of Interspecific Jumping

**DOI:** 10.3390/v17101320

**Published:** 2025-09-29

**Authors:** Caty Martínez, Daniel Echeverri-De la Hoz, Alfonso Calderón, Yésica López, Camilo Guzmán, Ketty Galeano, Valeria Bertel, Bertha Gastelbondo-Pastrana, Salim Mattar

**Affiliations:** 1Instituto de Investigaciones Biológicas del Trópico, Universidad de Córdoba, Montería 230001, Colombia; catymilenam@correo.unicordoba.edu.co (C.M.); dan.echeverri@mail.udes.edu.com (D.E.-D.l.H.);; 2Programa de Bacteriología y Laboratorio Clínico, Facultad de Ciencias Médicas y de la Salud, Universidad de Santander, Valledupar 520002, Colombia; 3Departamento de Bacteriología, Facultad Ciencias de la Salud, Universidad de Córdoba, Montería 230001, Colombia

**Keywords:** zoonosis, public health, genomic surveillance, chiroptera, one health

## Abstract

Since the appearance of the Severe Acute Respiratory Syndrome (SARS) virus, there has been increased interest in understanding the role of bats in the maintenance and circulation of coronaviruses. This study aimed to describe the phylogenetic and evolutionary relationships and antigenic architecture of a new coronavirus detected in bats in the Department of Córdoba. In a surveillance study of pathogens of interest to public health, a bat *Phyllostomus hastatus* was captured. Rectal swabs samples were collected from the bats, and RNA was extracted and sequenced using NGS with MGI-G50 equipment. The results were analyzed using bioinformatics software. A contig of 28,619 nucleotides associated with the Coronaviridae family was obtained. Phylogenetic and molecular clock analyses of the ORF1ab gene revealed a novel divergent Alphacoronavirus that originated directly from an ancestral node. The analysis of the spike (S) protein and receptor-binding domain (RBD) is similar to that of humans (HCoV-229E) and porcine coronaviruses. In silico analysis suggests potential RBD interaction sites with human and pig cellular receptor aminopeptidase N. There is a possible risk of interspecies jumping of the new AlphaCoV/*P. hastatus* in humans and pigs. This is the first study to perform phylogenetic, evolutionary, and antigenic characterization of bat coronaviruses in Colombia.

## 1. Introduction

Bats have been identified as reservoirs of multiple pathogens that can be transmitted to intermediate hosts and humans owing to the destruction of their natural habitats and increased ecological relationships [1]. The emergence of zoonoses is a consequence of the interaction of environmental, social, and economic factors, which arise at the interface between animals and humans and have significant impacts on public health and the economy, as in the case of the SARS-CoV-2 pandemic [2].

Owing to genetic and evolutionary evidence that coronaviruses (CoV) are pathogenic to humans, Alphacoronavirus and Betacoronavirus that infect mammals have been widely studied in bats [3,4]. Betacoronavirus, including SARS, MERS, SARS-CoV-2, HKU-1 and OC43 are important to public health, and Alphaconaviruses, including HCoV-229E and HCoV-NL63, cause colds [5,6].

The search for CoVs in bats is crucial for early detection to understand eco-epidemiology and implement strategies to prevent possible viral spillover. In the Americas, CoVs have been reported in bats in the United States, Brazil, Panama, Argentina, Peru, Mexico, Costa Rica, and Colombia [7]. Although the identified sequences were grouped with Alphacoronavirus, these findings indicate the need for a zoonotic risk assessment and continued surveillance of CoVs in New World bats. CoVs are characterized by their large, linear, monopartite ssRNA (+) genome, which allows them to have additional plasticity to accommodate and modify genes. A frequency of homologous RNA recombination has been described, which causes mutations that can introduce new properties and increase inter-species jumping, facilitating the spread to new hosts [5,8].

The ability of CoVs to enter a host cell is mediated by the spike protein (S) through its receptor-binding domain (RBD), followed by replication and subsequent transmission of viral particles [9]. The mechanisms of viral dispersion are related to respiratory particles, aerosols, direct contact, and fecal-oral transmission, and the clinical manifestations of CoV infections are characterized by respiratory and gastrointestinal tract conditions in different hosts [10]. There are 10,626 unique CoV sequences in bats worldwide, representing more than 45% of the viral databases available for these mammals (consulted 25 August 2025) [7]. Although only one-third of bat species have been tested, the diversity of bat CoVs is estimated to be greater [5]. Bats host various CoVs that extend across six continents [11]. CoVs can generate mutations when adapting to new hosts because potential zoonotic pathogens require ongoing management to achieve genetic characterization for timely detection and notification to public, human, and animal health surveillance systems.

This study aimed to describe the phylogenetic and evolutionary relationships and antigenic architecture of a new CoV detected in bats in the Department of Córdoba.

## 2. Materials and Methods

### 2.1. Capture and Ethical Aspects of the Study

Within the framework of an eco-epidemiological surveillance study of coronavirus in the department of Córdoba, Colombia, in 2022 [12], an adult male *Phyllostomus hastatus* was captured using mist nets (6 × 2) strategically located on a farm with a forested enclave in the municipality of Moñitos, Córdoba, Colombia (76°05′21″; 9°15′13″). In the reference study, the bats were taxonomically identified using dichotomous morphometric keys [13]. Rectal swabs and intestinal tissue samples were collected and stored at −80 °C. The study was approved by the Ethics Committee of the Faculty of Veterinary Medicine of the University of Córdoba (Act 003-06-12-2019) and the framework permit for the collection of specimens for non-commercial scientific research purposes from the National Environmental Licensing Authority ANLA (resolution 00914 of 2017).

### 2.2. Sequencing

Genetic material was extracted from the rectal and intestinal swabs. RNA was obtained using the Gene JET RNA purification kit (K0732 Thermo Fisher Scientific™ Waltham, MA, USA), and contaminating DNA was removed with DNAse I from Promega™ (Madison, WI, USA). Clean-up and purification were performed using the GeneJET RNA Cleanup and Concentration kit (Thermo Fisher Scientific™ Waltham, MA, USA). Subsequently, RNA concentration and integrity were determined using a Qubit™ fluorometer (Thermo Fisher Scientific™ Waltham, MA, USA). For metatranscriptomic sequencing, samples were processed using the Paired-End FCL 150 MGIEasy Fast RNA Library Prep Set™ (Shenzhen, China), which included RNA fragmentation into 250-nucleotide products, reverse transcription and second-strand synthesis, end-repair, and barcode ligation. Fragment size was assessed fluorometrically using an Agilent Technologies™ Bioanalyzer (Santa Clara, CA, USA). Sequencing was performed using high-throughput DNA nanobead (DNB) technology using the MGI-G50^®^ (Shenzhen, China).

### 2.3. Bioinformatics Analysis

The sequencing results were subjected to quality control and read filtering using the Fastp software v.1.0.1, which allowed the removal of low-quality reads and retention of sequences with an average Phred score above Q20. Fastp was used to verify the per-base quality distribution and ensured that only high-quality reads were obtained [14]. De novo assembly to obtain contigs with a minimum size of 300 nucleotides was performed using MEGAHIT software v.1.2.9 [15]. Taxonomic assignment was performed by comparison with the non-redundant (nr) protein database from GenBank using DIAMOND v0.9.25 in BLASTX mode, and taxonomic classification was performed with MEGAN6 6.25.9 [16]. Contigs of interest were further validated by comparison with the NCBI BLASTn and BLASTx databases of the NCBI platform, to confirm their viral origin [17]. The selected viral contigs were subsequently used as a reference genome for read mapping with Bowtie2 v.2.5.0, allowing alignment of the raw fastq data back to the assembled sequence [18]. The aligned reads were then extracted into a bam file for visualization in UGENE v50.0 [19] and sequencing depth using SAMtools 1.21 [20]. Genome annotation was performed using the Prokka 1.14.6 to predict coding sequences and assign functional information [21] (Appendix A).

### 2.4. Phylogenetic Analysis

Reference genomes of viruses important to animal and human health were downloaded from GenBank, and multiple alignments of the ORF1ab were performed using MAFFT v7.520 [22] and maximum-likelihood phylogenetic reconstruction (1000 bootstrap) using IQ-TREE v2.3.1 [23]. The substitution model was automatically selected by Model Finder, resulting in LG + F + I + G4, and the alignment was manually curated using UGENE v50.0. The tree was rooted based on the substitution rate measured over time by exploring the temporal signal in the data using the TreeTime 0.11.3 [24]. Accession numbers of the obtained sequences and all reference sequences used in the phylogenetic tree are provided (Appendix A). Subsequently, a comparison with reference genomes was performed to identify variations at the amino acid sequence level using SimPlot++V1.3 [25].

### 2.5. Evolutionary Analysis

The software Bayesian Evolutionary Analysis Sampling Trees (BEAST v2.7.4) was used to perform the molecular clock analysis using the ORF1ab gene sequences. Multiple sequence alignment were performed using MAFFT v7.520. The alignment was manually curated with UGENE v50.0. The parameters were set using BEAUti with a strict molecular clock under a Coalescent Constant Size approach and the substitution model GTR + Γ. The analysis was performed using four independent MCMC chains, each with a length of 100,000,000 generations and a sampling frequency of 10,000. The results were visualized using Tracer V1.7.2 [26]. An efficient sample size (ESS) > 200 was considered in all statistical analyses, as well as the convergence of the Markov chains [27]. The generated tree information was then summarized using TreeAnnotator with a 10% burn-in, and the tree with the highest posterior probability of all nodes was selected. Finally, the tree was visualized and edited using FigTree (https://tree.bio.ed.ac.uk/software/figtree/).

### 2.6. Analysis and Molecular Docking of S Protein

To recognize the different subunits of the S protein containing the RBD, the 3D structure of the obtained protein was generated using homology-based modeling. Amino acid sequences were used as inputs for the Swiss-Model server [28]. The selected template was the PDB structure 6jx7.1.A, which corresponds to the spike protein of an Alphacoronavirus. The model was refined and visualized using ChimeraX software v. 1.7.1. [29]. Physicochemical parameters, such as isoelectric point and molecular size, were analyzed using ProtParam, and functional domains were obtained using the PROSITE server [30].

The conserved domains of the spike protein were identified in the 3D model to determine the functional regions. The hypothetical active site was mapped in the 3D model using ChimeraX. Metals, charged and hydrophobic residues, pockets, and phosphorylation and glycosylation sites were considered. The active site was compared with proteins from phylogenetically related viruses to identify putative cellular receptors [31,32].

The crystallographic structure of the putative cellular receptor was obtained from the Research Collaboratory for Structural Bioinformatics Protein Data Bank (RCSB PDB) [31]. Specifically, porcine aminopeptidase N (pAPN) was obtained from PDB ID: 4F5C, and the human aminopeptidase N (hAPN) from PDB ID: 4FYQ. Before the docking analysis, a 3D homology model and crystallographic structure were prepared using ChimeraX by adding charges and hydrogen bonds. An information-based flexible docking approach was used to model biomolecular complexes using the HADDOCK 2.4 [33]. The models were evaluated using van der Waals energy values, the buried surface area, and the Z-score. Finally, the contact points were visualized and predicted using ChimeraX.

## 3. Results

### 3.1. Genomic, Phylogenetic, and Evolutionary Analyses

In a previous eco-epidemiological surveillance study of coronavirus in the department of Córdoba, Colombia, we captured 262 bats, 55 of which harbored the RdRp gene of CoV [12]. Meta-transcriptomic sequencing of total RNA from the rectal swab of Phyllostomus hastatus (Phyllostomidae) generated a viral contig of 28,619 nucleotides, which corresponded to 2.13% (367,234/17′207,820) of all reads. All bases were covered by at least one read, indicating 100% coverage and an average depth of > 200X (BioProject: PRJNA1162262- SRA: SRS23951049). Seven open reading frames (ORFs) encode structural proteins, a polyprotein that contains the machinery for viral replication, and three hypothetical proteins (Appendix A). Using a maximum likelihood phylogenetic tree with reference sequences in GenBank, the viral genome was designated as Alphacoronavirus, corresponding to the ORF1ab gene. This new sequence was designated as AlphaCoV/*P. hastatus* (Figure 1A). Additionally, the new AlphaCoV/*P. hastatus* was compared with 72 sequences of the genus Alphacoronavirus detected in bats from different areas (Figure 1B). AlphaCoV/*P. hastatus* is located in the South American clade, which is close to the sequences from bats in Brazil and Peru.

A comparison of the identity patterns of AlphaCoV/*P. hastatus* genome with those of other Alphacoronavirus HCoV-229E, HCoV-NL63 and Rhinolophus HKU10, revealed similarities of approximately 60%. In contrast, similarities with the Betacoronavirus MERS, SARS and SARS-CoV-2 were less than 40% (Figure 2).

The evolutionary history of the new AlphaCoV/*P. hastatus* was determined using Bayesian analysis of ORF1ab gene with four independent MCMC chains, each with a length of 100,000,000 iterations and a sampling frequency of 10,000. Thus, it was possible to establish that the root of all the sequences used dates back to approximately 31,000 years, and the most recent common ancestor (TMCRA) for the node that gave rise to the new Alphacoronavirus was approximately 11,000 years (Appendix A).

### 3.2. Antigenic Modeling of the Spike (S) Protein of AlphaCoV/P. hastatus

Conserved domains of Coronaviridae were identified from the amino acid sequence encoding the S protein. The architecture of the S protein of AlphaCoV/*P. hastatus* was identified (Appendix A). The spike protein was characterized using computational modeling to evaluate its physicochemical parameters. The generated trimer was structurally similar to that described for the Coronaviridae family (Figure 3). It was compared with the receptor-binding domain of other Alphacoronavirus of importance to human and animal health (Appendix A). Analyses showed the homology modeling of AlphaCoV/*P. hastatus* S protein concerning the S protein, transmissible gastroenteritis virus (TGEV), and human CoV HCoV-229E.

Analysis of homologous structures (TGEV and HCoV-229E) showed a conformational similarity in the S1 subunit, mainly in the C-terminal domain. In this region, the RBD generates a receptor-binding edge that comprises two interaction loops. Tyr is primarily located between the β1 and β2 sheets at the first binding edge. The second loop was located between the β6 and β7 sheets of AlphaCoV/*P. hastatus* and HCoV-229E, where they maintain a hydrophobic amino acid, specifically *Leu* and *Val*, respectively. In TGEV, a second loop is formed between the β3 and β4 sheets, maintaining an aromatic amino acid (*Trp*). In addition, unlike HCoV-229E and TGEV, AlphaCoV. The AlphaCoV/P. hastaus region presents a structure dominated by loops and not β-sheets, which could be related to the connectivity and flexibility of the protein to bind to the receptor (Appendix A), as previously described.

### 3.3. Molecular Docking

Hypothetical active site established for AlphaCoV/*P. hastatus* was evaluated to predict the contact points with the cellular receptors porcine aminopeptidase N (APN) and human aminopeptidase N (APN). For the in silico assay, AlphaCoV/*P. hastatus* RBD-porcine APN complex, the interaction showed a Van der Waals energy of −69.3 and a contact surface of 2287.1 Å2, where five residues involved in forming hydrogen bonds were identified. The proposed binding edge (RBD tip) came into contact with the distal region of the APN ectodomain membrane without the generation of hydrogen bonds. However, an interaction was observed between the residues Glu731 and Trp737 located in the APN side chains (Figure 4A). Molecular Docking of AlphaCoV/*P. hastatus*. The hRBD-APN complex showed a contact area (2989 Å2) and van der Waals force (−88.6). Nine residues of AlphaCoV/*P. hastatus* RBD is involved in the human RBD-APN complex. Furthermore, the residues His533, Thr535, Asp537, Thr565, and Thr569 formed bonds with high specificity and strength (Å2 > 1500) (Figure 4B). In both assays, His533 and Thr535 were involved in the interaction with A.

## 4. Discussion

This study describes the first complete genome of a new CoV in Colombia and South America, found in the bat *Phyllostomus hastatus* from the Colombian Caribbean. The *P. hastatus* bat is one of the largest bats in Latin America and has been reported in Honduras, Brazil, and northern Argentina [34]. In Colombia, it is distributed throughout the territory between 0 and 2000 m above sea level [35]. It is an omnivorous species with gregarious habits, cooperative behaviors, and parental care, with colonies of more than 20 individuals [36].

These bats are found in tree trunks, caves, termite nests, and human buildings [35]. In these refuges, *P. hastatus* appears alone or associated with other species, such as *Carollia perspicillata*, *Desmodus rotundus*, *Saccopteryx bilineata*, *Trachops cirrhosus*, *Noctilio albiventris*, *Uroderma bilobatum*, *Artibeus lituratus*, and *Molossus molossus* [37]. The sympatric association between different Chiroptera species facilitates CoV transmission [38].

Bat-CoV diversity is correlated with the taxonomic diversity of the host; therefore, the highest number of Bat-CoV species is expected to be found in areas with the highest levels of bat species richness [39]. Among the 1400 species of bats worldwide, Colombia has 222 (16%), making it the Latin American country with the greatest diversity in this taxonomic group [40]. However, to the best of our knowledge, this is the first report of CoV infection in bats in Colombia.

Alphacoronavirus has been reported in the Phyllostomidae family, and two of the 28 *Phyllostomus* genera have been found in *P. hastatus* [7]. However, these are partial sequences of the RdRp gene. Therefore, the sequence presented in the present study is the first complete genome of Alphacoronavirus in the *Phyllostomus* genus from America and the first genetic characterization of CoVs in bats from Colombia. In the department of Córdoba, partial sequences of the RdRp gene of AlphaCoV have been reported in different species and localities; therefore, the present study shows the genomic and antigenic characterization of an AlphaCoV that could be circulating in the region [12].

Identity patterns of AlphaCoV/*P. hastatus* genome allows the assessment of phylogenetic relationships with Bat-CoVs circulating worldwide [11]. They demonstrated a close relationship with bat CoVs in America. Simultaneous circulation has been observed in the South American clades of Brazil, Peru, and Colombia [41,42,43]. Another clade was located in other sequences of *AlphaCoV*, *Tadarida brasiliensis*, reported in Argentina, similar to those described in the Old World.

The evolutionary association between CoVs and bats is well established [3,5]. Since the Miocene, the Phyllostomidae family has greatly adapted to the Neotropical environment [44]. In this study, the ancestor of American Alphacoronavirus was identified in the Pleistocene. This era coincides with environmental changes, such as glaciation, which may generate selective pressures favoring Alphacoronavirus [5].

Evolutionary estimates of AlphaCoV/*P. hastatus* indicates the phylogenetic node from which it originated approximately 11,000 years ago, and it is unknown whether the CoVs of Neotropical bat populations have changed hosts. Inter-species jumps are believed to occur more frequently in Alphacoronavirus than in Betacoronavirus [4]. Therefore, the analysis of similarities between CoVs is important in public health because it allows us to understand the ecology of the viruses, their genetic diversity, and the molecular mechanisms underlying potential zoonotic infections [3].

Previous studies have indicated that important CoV recombination events occur in bats [45,46], especially in genes associated with the spike (S) protein, which is responsible for entry into the host cell through its receptor-binding domain (RBD) [45,47]. Therefore, comparative analyses of the spike proteins of different Alphacoronavirus have provided insights into their transmission capacity and possible interspecies jumps [5].

Results for AlphaCoV/*P. hastatus* showed similarities at the RBD level with porcine CoV TGEV and human HCoV-229E. Conserved residues represent approximately 30% of the amino acids in this region of the protein. Structurally, position 520–660 of the spike protein of AlphaCoV/*P. hastatus* completely resembles the active site of the protein and contains all three receptor-binding sites [48,49]. Additionally, similar physicochemical features indicate thermal stability and hydrophobicity [48].

Alphacoronavirus HCoV-229E and TGEV use cell surface aminopeptidase N (APN), a membrane-bound metalloprotease, as a cellular receptor [50]. Therefore, the in silico assays developed here were with the APN receptor, which interacts with the viral RBD to enter host cells. Analyses of AlphaCoV/*P. hastatus* structure hastatus RBD, compared to the RBDs of HCoV-229E and TGEV, are based on the ability of the RBD to acquire new interactions with receptors [51].

The evolutionary origin of HCoV-229E in bats [52] explains the similarities in RBD levels of AlphaCoV/*P. hastatus*, where a hydrophobic amino acid is maintained in the second loop, and its structure is based mainly on loops and short β-sheets, allowing greater flexibility for binding to the APN receptor. However, in silico analysis indicated that the new AlphaCoV binds to subunit IV but not to subunit II of human APN, unlike HCoV-229E [51].

The exposure of APN subunit IV may vary in different tissues and assemblies with the RBD of AlphaCoV/*P. hastatus* is possible. In the case of HCoV-229, multiple structural changes and replacements of the RBD have been reported over time [51]. This demonstrates the adaptability of Alphacoronavirus with respect to specificity (cell tropism), which could affect its transmissibility and zoonotic potential [53].

Regarding human-bat interface activities and local exposure risks, a potential spillover should be considered. Although no cases of CoVs transmitted by bats to humans have been reported in Colombia, the possibility of viral spread to humans should be considered because of several factors. First, bat habitats are associated with agricultural and urban areas that are affected by unplanned growth. In addition, varied feeding habits and a high prevalence of CoVs in bats of the Phyllostomidae family have been recorded in this study area [11]. Considering the ecological factors and human-bat interphase, ecosystems altered by humans can stress bats and promote viral spillover [54]. Transmission of AlphaCoV/*P. hastatus* from bats to pigs has been previously described as a risk because pigs appear to be a host that amplifies the viruses [55].

Similarities between AlphaCoV/*P. hastatus* RBD with transmissible gastroenteritis virus (TGEV) were evident in the C-terminal region of S1. At this site, the conformation of the RBD-binding loops and the Tyr residue in the first loop (β1-β2) were maintained. In TGEV, the Trp residue is present in the loop (β3-β4), which interacts with subunit IV of porcine APN and promotes π-stacking interactions and hydrogen bonds [56]. Additionally, the spike protein of AlphaCoV/*P. hastatus* interacts with the Glu731 and Trp737 residues of APN, contributing to the receptor-binding specificity of TGEV [56]. The present study has limitations in computational predictions; therefore, experimental validation is needed to determine whether the virus can successfully infect cells of these species and cause disease.

There is no recent evidence of TGEV circulation in Colombia; however, there is evidence that Alphacoronavirus causes porcine epidemic diarrhea [49,57]. Studies on CoVs that affect pig production have shown a high rate of recombination at the RBD level, which impacts membrane fusion processes and confers resistance to virus neutralization [58]. Genome analyses, especially those of the S protein and its receptors, are required to understand the interspecies jumps at the molecular level [8]. Furthermore, information on livestock production systems is crucial for developing new vaccines and drugs.

Health authorities (WHO-OIE) suggest maintaining surveillance at the ecological interface between bats and pigs [55], given the indicative events of transmission between species, such as that reported for the CoVs SADS-CoV affecting pig production in China, where genetic similarity with HKU2 found in Rhinolophus bats was demonstrated [59]. In this context, previous zoonotic events involving RNA viruses have shown that their evolution and successful transmission to humans often require an intermediate or amplifying host, such as pigs [60].

## 5. Conclusions

The present study describes the first complete genome of a new CoV found in the bat *Phyllostomus hastatus* in Colombia and South America. Structural analysis in silico of the S protein suggests that there is a similarity in the receptor-binding site (RBD) of the S protein of porcine and human CoVs, with the cellular receptor aminopeptidase N being shared by vertebrate species, which could represent a potential risk of interspecies jumps. The findings of this study lay the foundation for surveillance using a comprehensive One Health approach.

## Figures and Tables

**Figure 1 viruses-17-01320-f001:**
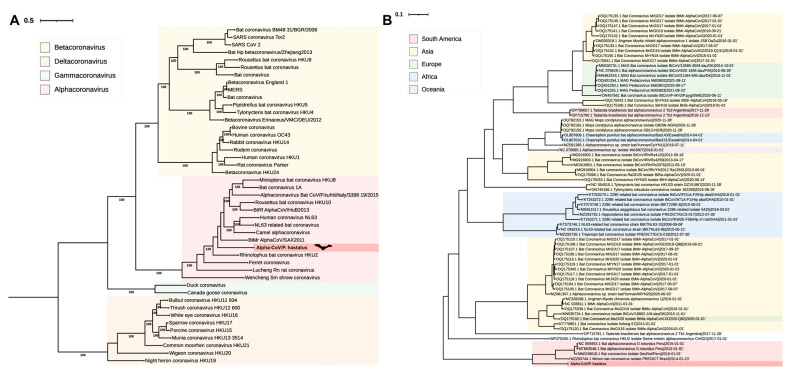
Maximum likelihood phylogenetic tree of the Coronaviridae family (**A**) and Alphacoronavirus genus (**B**). The phylogenetic tree of bat Alphacoronavirus was time-calibrated with Treetime under the rerooting and substitution rate estimation options. The branch scale represents the number of amino acid substitutions in ORF1ab, and the numbers at the nodes indicate the statistical support (bootstrap: 1000). AlphaCoV/*P. hastatus* (red).

**Figure 2 viruses-17-01320-f002:**
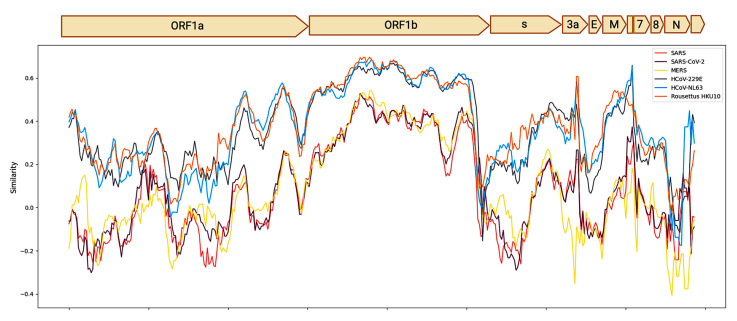
Genome identity patterns of AlphaCoV/*P. hastatus* consensus sequence and Coronaviruses of public health importance.

**Figure 3 viruses-17-01320-f003:**
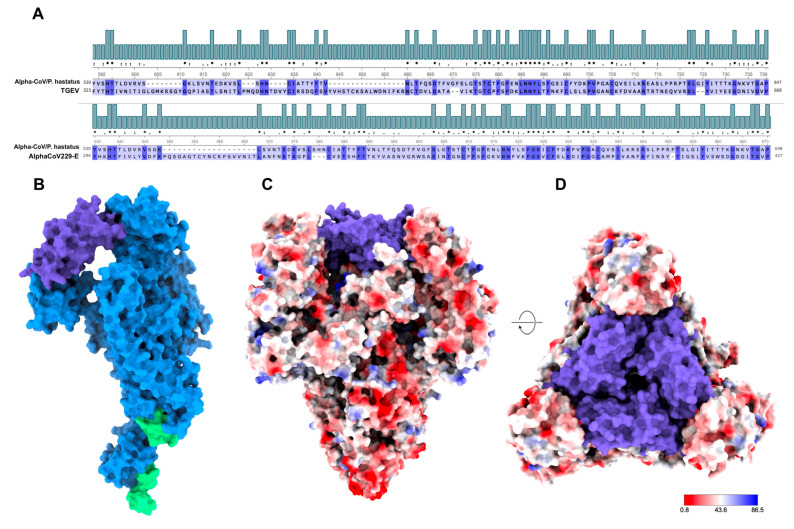
Homology modeling of AlphaCoV/*P. hastatus* S protein. (**A**). Alignment of the AlphaCoV/*P. hastatus* of TGEV and HCoV-229E. The region shown corresponds to the RBD of the spike protein. The conserved residues are shown in purple. (**B**). 3D structure of the S protein monomer; functional domains of the protein are represented in green; proposed RBD for Alpha-CoV/*P. hastatus* are indicated in purple, and the conserved regions are shown in blue. (**C**,**D**). 3D structure of the S protein trimer from the side and top views; proposed RBD for AlphaCoV/*P. hastatus* are shown in purple. The color scale represents the isoelectric point of the protein. The molecular size was 155.45 kDa, isoelectric point was 6.169, hydrophobicity was 0.032, and Aliphatic index 83.97. ‘*’ complete amino acid identity; ‘:’ substitutions between residues with similar properties; ‘.’ semi-conservative substitutions between residues with similar properties.

**Figure 4 viruses-17-01320-f004:**
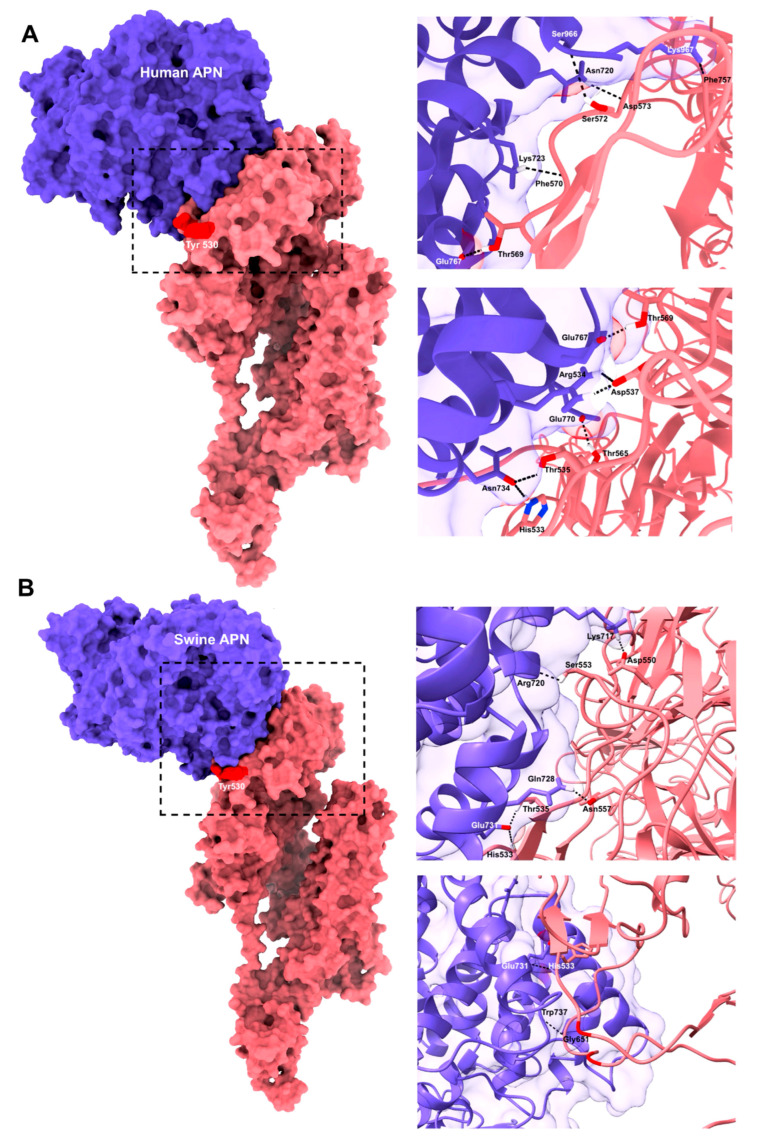
(**A**). Molecular Docking of AlphaCoV/*P. hastatus* RBD-human APN complex. Spike protein monomers are bound to APN. The contact region between the two proteins is indicated. Binding details between AlphaCoV/*P. hastatus* RBD residues and human RBD-APN. The residues Thr535 and His 533 of the RBD remain active amino acids that bind to Asn734 of human APN. (**B**). Molecular Docking of AlphaCoV/*P. hastatus* RBD-APN porcine Monomer of the spike protein bound to APN. The contact regions between the two proteins are shown. Details of the binding between the residues of AlphaCoV/*P. hastatus* RBD, and porcine RBD-APN. The binding of the amino acid Glu731 to Thr535, His 533, and Trp737 to Gly651 of the RBD was maintained. The purple regions correspond to the indicated cell receptor, human or pig, and the pink regions correspond to those belonging to the AlphaCoV/*P. hastatus* RBD.

## Data Availability

The datasets generated during the current study are available in Genbank (SRA data-BioProject ID: PRJNA1162262, Accession: SAMN46506513 ID: 46506513).

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
