# Peer review of "New Coronavirus in Colombian Caribbean Bats: In Silico Analysis Reveals Possible Risk of Interspecific Jumping"

_viruses, 2025, doi:10.3390/v17101320_

Round 1
Reviewer 1 Report
Comments and Suggestions for Authors
From my perspective, the study is relevant to the Viruses readership and contributes to essential surveillance efforts aimed at detecting and understanding the emergence of novel pathogens.
Overall, the study is well designed and clearly relevant in terms of public health. It provides a comprehensive analysis of a new strain of Alphacoronavirus, ranging from sequencing to docking with receptor proteins. What I particularly appreciate about this work is the cross-disciplinary nature of the analyses. Few studies bring together such a wide range of bioinformatics expertise: NGS, maximum likelihood phylogeny, Bayesian approach, structural inference, and docking.
However, I believe that verbatim of methods would benefit from clearer & deeper discussion and I'm a bit concern about the potential for chimeric misassembly, which can always occurs in metagenomics datasets & should be explicitly refuted in the body of the manuscript.
See attached file for complete review.

Reviewer 2 Report
Comments and Suggestions for Authors
Dear Authors!
Recently, there has been increased interest of new viruses investigating in poorly studied reservoirs. Bats are one of these reservoirs. Also, investigating new regions is important.
Your article is interesting and provides new knowledge about coronaviruses circulation.
I have some comments:
Line 40 – HKU-1 is betacoronavirus. If you mention all human coronaviruses in Introduction, you should also mention OC43.
Lines 57-58 – Also, it is better to add the web resource reference, indicating request date. I see different number of sequences in the Database (11,206 sequences and 43%).
Lines 59-60 - You used reference to 2021. I recommend to find modern article.
Why do you use different names for same virus (AlphaCov/P.hastatus; Alpha Cov Colombia (graphic in Figure 3); Bat_CoV-Phillostomus_hastatus_Colombia|2021-06-16 (graphic in Figure S2))?
Paragraph 2.1 states that material was collected in 2022. Figure S2 shows that virus is from 2021-06-16. Where is right place?
Lines 257-258 - You used reference to 2017. I recommend to find modern article.
Lines 528-529 – It needs to correct reference style.
Reviewer 3 Report
Comments and Suggestions for Authors
The authors describe the full-genome detection of a novel alphacoronavirus in the bat Phyllostomus hastatus in the Colombian department of Cordoba and performed in silico phylogenetic, evolutionary, and structure-antigen analysis. A contig of 28619 nt in length with full coverage (coverage over 200) was collected, the virus was assigned to Alphacoronavirus; the RBD of the S-protein is similar to HCoV-229E and TGEV, and docking with human and porcine aminopeptidase N receptor was modeled. Raw and metadata are available. Conclusion: there is a possible risk of interspecies transmission (human/pig).
Pros:
This is the first complete CoV genome from bats in Colombia/South America and the first for the genus Phyllostomus in the Americas.
Nearly 100% coverage, average depth over 200; viral read fraction ~2.1% - sufficient for good consensus sequencing.
Project/sample identifiers in the SRA are provided.
Cons:
All conclusions about spillover tropism/risk are based on modeling; there is no PCR screening of other individuals for this genome, no cellular experiments (pseudovirus/APN binding), and no serology. The wording is in places overly categorical for in silico work.
One sample and one consensus is shown without validation by repeated sequencing/primers, without analysis of intra-household variability, and without assessment of contamination.
Instruments/parameters are confused in places (e.g., mention of “chain of 400,000,000 iterations” next to TreeTime, which is not MCMC), typos and variation in nomenclature (AmpN/APN; TGEV/TGVE; MAAFFT instead of MAFFT; ‘Alphacoronavirus’/“Alphacaronavirus”, etc.).
It is written that the study is “in accordance with the Declaration of Helsinki”, which refers to research involving humans - inappropriate for wild fauna; references to ANLA permits and local committee approval are sufficient.
What can be quickly corrected (minor):
Proofread text for typos, unify nomenclature: APN (not AmpN), TGEV, MAFFT, “Alphacoronavirus”, units, commas/dots in numbers, harmonization of protein/gene names.
Soften language about “risk of interspecies transfer” to the level of hypothesis/in silico indication, add an explicit disclaimer about the need for experimental validation (hAPN/pAPN pseudovirus test, BLI/SPR for binding.
Clearly separate TreeTime (ML dating) and BEAST (MCMC molecular clock).
Perhaps attach an additional table with mapping statistics: total number of reids, proportion viral, average/median depth across the genome, areas of low coverage.
Give PDB-ID of templates for homology, GMQE/QMEAN for models.
Fix block about ethics: remove “Declaration of Helsinki”, leave permit/act numbers for animals.
Recommended (optional) changes, which will greatly improve this work:
Alternative assembly with dif. parameters, control of possible chimerism, validation of S-gene by RT-PCR and Sanger/amplicon NGS.
Minimally: screening of available samples (from the same mice) with primers to the new virus to assess prevalence and variability.
Pseudovirus with S of the new CoV on cells expressing hAPN/pAPN, which will significantly strengthen the section on interspecies risk.
The paper is valuable for its new sequence and good in silico analysis, but requires methodological cleaning, language revision, and, if possible, minimal experimental validation of key biological conclusions. After elimination of these remarks, publication is recommended.
Round 2
Reviewer 3 Report
Comments and Suggestions for Authors
Accept in present form